# The impact of social media interventions on eating behaviours and diet in adolescents and young adults: a mixed methods systematic review protocol

Hao Tang  , Marie Spreckley, Esther van Sluijs, Amy L Ahern, Andrea D Smith

ALA and ADS are joint senior authors.

MRC Epidemiology Unit, University of Cambridge School of Clinical Medicine, Cambridge, UK

**Correspondence to**
Hao Tang;
Hao.tang@mrc-epid.cam.ac.uk

## ABSTRACT

**Introduction** Adolescents and young adults are susceptible population when it comes to healthy eating and dietary behaviours. The increasing use of social media by this age group presents a unique opportunity to promote healthy eating habits. Social media has become a popular platform for promoting health interventions, particularly among young people. However, there is a lack of consensus on the effectiveness of social media interventions in this population. This mixed-method systematic review aims to synthesise the available evidence on the impact of social media interventions on healthy eating behaviours among young people, their qualitative views and user experiences, and the intervention characteristics, behaviour change theories and techniques used to promote healthy eating.

**Methods and analysis** We will conduct a comprehensive search of seven electronic databases, including ASSIA, Cochrane Library, Embase, MEDLINE, PsycINFO, Scopus and Web of Science. The search strategy will use a combination of Medical Subject Headings terms and keywords covering three domains: social media, eating behaviours and young people. The search will be limited to peer-reviewed published papers in any language, published from 2000. Three independent reviewers will screen studies based on predetermined eligibility criteria. Data will be extracted and analysed using a convergent segregated mixed-method approach. We will use random-effect meta-analysis or Synthesis Without Meta-analysis for quantitative data and thematic synthesis for qualitative data. Finally, narrative synthesis using concurrent triangulation will be used to bring together the results of the mixed-method data analysis to provide a comprehensive and integrated understanding of the impact and other features of social media interventions. This systematic review will adhere to the Preferred Reporting Items for Systematic Reviews and Meta-Analyses.

**Ethics and dissemination** Ethical approval is not required since this systematic review will not collect original data. The outcomes of this review will be shared through peer-reviewed publications and conference presentations and will contribute to the PhD thesis of the primary author.

**PROSPERO registration number** CRD42023414476.

## STRENGTHS AND LIMITATIONS OF THIS STUDY

⇒ Utilisation of a mixed-methods approach to incorporate both quantitative and qualitative data for a comprehensive evaluation.
⇒ Examination of behaviour change techniques within interventions offers insights into underlying mechanisms of action.
⇒ Inclusion of a broad range of eating behaviour outcomes, including disordered eating behaviours, enhances the scope of potential findings.
⇒ Incorporation of studies published in multiple languages helps capture diverse populations and cultural perspectives.
⇒ The interpretation of quantitative results may be limited by the lack of standardised reporting on eating behaviours and the small final number of eligible studies.

## INTRODUCTION
### Rationale

Developing behaviour change interventions that effectively promote healthy eating habits and positive attitudes to food among adolescents and young adults is a critical public health priority. This age group experiences significant physical, psychological and social changes, making it an important developmental period to support and establish healthy eating behaviours that have the potential to persist into later adulthood.[1–3] However, the majority of adolescents and young adults do not meet the recommended healthy eating guidelines[4 5] This can have negative consequences for physical and mental health. Suboptimal dietary intake in this age group is linked to elevated risk for the development of chronic diseases such as type 2 diabetes, cardiovascular disease and certain types of cancer later in life.[6–8] At the same time, the present food environments facilitate access to palatable foods high in saturated fats, salt and

free sugar.[9] These often calorically dense foods may be highly processed and widely promoted via sophisticated marketing strategies, often online, targeted at young people.[10 11] A report by WHO estimated that 340 million adolescents (10–19 years of age) were living with obesity in 2022.[12] Even in youth, obesity directly increases the risk for various non-communicable diseases and indirectly contributes to psychological distress.[13 14] Interventions are warranted that can support healthy eating habits among adolescents and young adults given the far-reaching benefits for both individuals and society.

The strong integration of social media into young people's lifestyles and daily behaviours presents a unique opportunity to transform contemporary approaches to support healthy eating.[15–19] Social media are internet-based applications that allow users to create a unique profile and enable passive and active exchange of text, image or video-based content, although specific features vary by platform.[20] Recent evidence suggested that most young people aged 13–35 years actively use social media. For example, 97% of 16–24-year-olds in the UK use social media on a weekly basis.[21 22] Social media platforms are also seen as trusted resources for information, with 52% of young people using social media to seek advice on health issues.[23] Although these platforms provide opportunities for people to engage with each other in ways that are beneficial, social media can also allow misinformation to proliferate, especially in relation to diet and other well-being-related behaviours. Research has shown that social media platforms can pose risks to young people via exposure to inaccurate or inappropriate content,[24 25] increase the risk for disordered eating behaviours like binge eating and restrictive eating[26–29] and are associated with health risk behaviours including alcohol, vaping and unhealthy dietary behaviours.[30]

It is also important to acknowledge that the global shift towards online platforms during the COVID-19 pandemic has had a significant impact on the way health information is delivered and received.[31 32] This trend has resulted in an increased acceptance of receiving health advice and interventions through social media platforms.[33 34] Furthermore, social media platforms have become more sophisticated, allowing for personalised and interactive content that can be tailored to individual users' needs and preferences. This, in turn, has the potential to increase engagement and improve the effectiveness of interventions aimed at promoting healthy eating behaviours. Given these developments, there is a need for updated research to assess the potential of social media interventions as a means of improving dietary behaviours and preventing diet-related diseases in adolescents and young adults.

Social media interventions can be defined as interventions that are conducted mostly through social media platforms (eg, Facebook, Instagram, YouTube, etc). Despite the increasing popularity of social media interventions to promote healthy eating habits and diets among adolescents and young adults,[35] a comprehensive overview literature is lacking. This will help consolidate the existing evidence on their effectiveness and address several research gaps, especially regarding the exploration of unintended consequences of providing such information online in a digital format. Previous systematic reviews in adjacent topics have mostly focused on the quantitative assessment of the effects of social media interventions on healthy eating behaviours among young people.[35 36] Important questions remain on the 'how' and on the design of the most beneficial delivery approach.[17 37 38]

There is a limited understanding of the strategies and mechanisms behind how social media influences young people's eating behaviours and diet and how these processes can be leveraged to promote healthy eating. To our knowledge, the most recent reviews that analysed the behaviour change techniques (BCTs) and content of social media interventions for eating behaviours only included papers from before 2014 only.[39] Systematically reviewing BCTs and explicitly reporting their application in the context of interventions are essential to understand the mechanisms of effectiveness and guide future intervention design.[40]

This mixed-method systematic review seeks to synthesise existing quantitative and qualitative research to assess the influence of social media interventions on healthy eating among young people, examining their experiences, perspectives and preferences. The goal is to identify effective intervention components—'active ingredients'—and to understand user satisfaction and long-term efficacy. Additionally, the review will address potential adverse effects, such as weight stigma and disordered eating, with the aim of informing the development of future interventions that balance positive impact with the minimisation of negative outcomes.

## OBJECTIVES

This mixed-method systematic review will evaluate existing quantitative and qualitative evidence on social media interventions for healthy eating in young people.

To address this overarching aim, we will:

1. Synthesise quantitative estimates of how social media interventions impact healthy eating behaviours among young people (13–35 years old)
2. Explore qualitative views and user experiences of young people who have participated in social media interventions aimed at promoting healthy eating behaviours
3. Identify the intervention characteristics, behaviour change theories and BCTs employed in social media interventions targeting healthy eating behaviours among young people (13–35 years old)

## METHODS AND ANALYSIS

The reporting of this systematic review will adhere to the Preferred Reporting Items for Systematic Reviews and Meta-Analyses.[41] This protocol has been registered in the

International Prospective Register of Systematic Reviews (ID: CRD42023414476). The study is planned to begin on 1 March 2023 and end on 1 May 2024.

## Literature search strategy

The information sources and search strategy for this mixed-method systematic review will be designed to ensure a comprehensive and systematic search of the available evidence. The following seven databases will be searched:

► ASSIA
► Cochrane Library (CENTRAL)
► Embase
► MEDLINE
► PsycINFO
► Scopus
► Web of Science

The search strategy will be based on a combination of relevant Medical Subject Headings terms and keywords covering three key domains (1) social media AND (2) eating behaviours AND (3) young people. The search terms are listed in online supplemental table S1. The search strategy will be iteratively tailored to the specific database to ensure that all relevant studies are identified while avoiding the inclusion of duplicates or irrelevant studies. The search was restricted to papers published after 2000 since most social media platforms used today were developed since then. The search will be conducted without language restrictions to ensure that all relevant studies are included in the review. Only peer-reviewed published papers will be included. A final search syntax for each electronic database is included in online supplemental table S2.

In addition to the databases, the reference lists of eligible studies and relevant systematic reviews and meta-analyses will be searched for additional studies that may have been missed in the initial database search. The searches will be rerun prior to the final analysis.

## Eligibility criteria

Study selection will adhere to the criteria briefly summarised in table 1 and detailed in online supplemental tables S3 and S4.

## Study designs

We will include quantitative, qualitative and mixed-method studies. Eligible quantitative studies will include experimental and observational studies, such as randomised

**Table 1** Inclusion and exclusion criteria

| Inclusion criteria | Exclusion criteria |
|---|---|
| Study design | |
| ► Quantitative, qualitative or mixed-method studies<br>► No restriction in language of studies or study. Original peer-reviewed primary research articles<br>► Year of publication: 2000 onwards | ► Non-peer-reviewed grey literature and unpublished trials<br>► Literature review (eg, scoping review and systematic review) |
| Population | |
| ► Age: 13–35 years old<br>► Can be included with OR without health conditions as long as the condition does not substantially influence diet/eating behaviours | ► Institutional populations<br>► Target participants with pre-existing health conditions<br>► Pregnant or lactating individuals |
| Intervention | |
| ► Researcher-led behavioural interventions or larger-scale behaviour change campaigns that were mostly conducted on commercial social media platforms | ► Interventions in clinical settings |
| Comparison | |
| ► Not necessary to have a control group for interventions | |
| Outcomes | |
| **Only include the paper when primary outcomes are included**<br>► **Primary outcome**<br>Quantitative: changes in diet/dietary behaviours/diet pattern<br>Qualitative: views and experience in relation to participating in an intervention aimed at changing diet/dietary behaviours/diet pattern<br>► **Secondary outcomes:** energy balance-related outcomes/behaviours such as physical activity, body weight or BMI | ► If the intervention only reported on outcomes that are not listed as the primary outcome, such as alcohol intake, smoking, nutrition supplement use, changes in knowledge about and attitudes towards eating and so on |

BMI, body mass index.

controlled trials (RCTs), cluster RCTs, quasi-randomised trials, before and after (pre-post) studies, and prospective and retrospective studies. We will also include qualitative studies that applied focus groups or one-on-one interview approaches. Only original peer-reviewed primary research articles will be included. No restrictions on language will be set, and only studies published after 2000 will be included. This is to ensure the systematic review captures the most relevant data within the context of the contemporary social media landscape and its potential to impact eating behaviours and diet of adolescents and young adults.

## Participants

The population of interest is adolescents and young adults between 13 years and 35 years of age. The eligible age range was chosen based on the following: (1) the lower limit of the defined eligible age range is 13 years because this is the minimum age for setting up an account on commercial social media platforms (eg, YouTube, Instagram, Twitter and TikTok); (2) 13–35 years comprises the highest percentage of active social media users among all age groups in many countries, such as the UK and the USA[42–45]; and (3) this age range includes important and formative developmental periods when individuals experience significant changes in their health status and social identities. In most countries, turning 18 years old is also the age they are legally considered an adult which brings along a new sense of autonomy as well as a wide range of social and legal obligations. With regard to physical health and behaviours, many individuals experience weight gain during the period between 18 years and 35 years of age.[46 47]

The present systematic review will exclude studies with a mean or median age reported to be below 13 years or above 35 years. Studies that exclusively involve participants with pre-existing health conditions that may influence dietary behaviours, such as type 2 diabetes, eating disorders, postsurgery, cancer or anaemia, will be excluded to ensure that the generalisability of the results is not biased by these conditions. Studies that selected participants based solely on a clinical outcome, such as allergy risk, may not be relevant to the research question and will also be excluded.

## Interventions

The scope of this systematic review will include both researcher-led behavioural interventions and larger-scale behaviour change campaigns that were mostly conducted on commercial social media platforms. These platforms include social media sites such as Facebook, Twitter, Instagram, TikTok and others that are profile based and allow for user-generated content sharing.[20] Social media platforms that are not classified as profile based with user-generated content will be excluded. For example, platforms like Zoom will be excluded as they are primarily designed for video communications. Similarly, social media platforms created specifically for an intervention will also be excluded as they may have unique characteristics that are not representative of commercial social media platforms. Where intervention delivery may rely on multiple components, only those interventions where the main component (or collectively the largest proportion of components) was delivered via social media will be included.

## Comparators

For eligible intervention studies, we will include studies with any type of comparator and those with no comparator (pre-post comparisons). For RCTs, comparators can include standard care/no intervention/waiting list and attention-matched controls that do not involve social media.

## Outcomes

The main outcomes of the quantitative studies will be changes in dietary behaviours and patterns as measured by questionnaires and/or diet diaries. For example, the measures of overall food consumption (including total daily calories and proportion of macronutrients); food variety, type and (sugar-sweetened) beverage consumption; diet quality indices; and other eating behaviours such as binge eating, restrictive eating, overeating and emotional eating. Studies must include at least one of these main outcomes to be included in the systematic review. Interventions only reporting on changes in knowledge about and attitudes towards cooking or changes in the frequency of home cooking and/or food preparation habits will be excluded.

For qualitative studies, the main reported outcome must include qualitative feedback, experiences and perspectives in relation to participating in a social media intervention that aims to change diet, dietary behaviours and/or diet pattern. This includes qualitative feedback on content/features, facilitators and barriers to using the interventions, preferences, expectations and feelings in relation to the social media intervention (or social media-delivered components).

Additional outcomes, when reported, will include energy balance-related outcomes such as physical activity, body weight and body mass index. These outcomes will provide additional information about the potential impact of social media interventions on the overall health and well-being of young people. Studies will be excluded if the intervention only reported on outcomes that are not listed as the main outcomes, such as alcohol intake, smoking or nutrition supplement use.

## Timing

Outcomes reported at all time points will be extracted. The primary analysis will focus on the outcomes that were collected and reported at the pre-defined end of the intervention.

## Settings

Only studies involving participants living in community-based settings will be included. We will exclude studies conducted in clinical settings.

## Data management and selection process

The data management and selection process for this review will be performed using a standardised and systematic approach. All studies identified through the search process will be imported into the Endnote X7 reference management software to remove duplicates. The remaining articles will be uploaded to Covidence (www.covidence.org), and more duplicates will be removed. Two independent reviewers will screen all study titles and abstracts in duplicate to determine their eligibility. Any discrepancies will be discussed and, when required, resolved through consultation with an independent reviewer. We will retrieve the full text of all articles identified as potentially relevant, and again, two reviewers will screen these to determine eligibility, with discussion to resolve discrepancies and consultation with a third reviewer where necessary. Reasons for exclusion of articles at the full-text screening stage will be recorded.

## Data items

Relevant data will be extracted from eligible full-text articles using a piloted and standardised data extraction form. Qualitative and quantitative data will be extracted and analysed separately. A subset of extracted and coded data (50% of studies) will be cross-checked by a second reviewer, and any discrepancies will be resolved in discussion with an independent third reviewer. The following items will be extracted from the eligible quantitative studies:

► General information (eg, study authors, publication year, country and funding source)
► Study details (eg, study aim, study design, randomisation method, blinding and allocation concealment)
► Participants information (eg, demographic characteristics, recruitment methods, sample size and health conditions)
► Attrition/adherence (eg, total number of participants at baseline and follow-up measurements, differential attrition, attendance, study withdrawal and lost to follow-up)
► Intervention information (eg, setting, content, BCTs, intervention duration and frequency, profession delivering the intervention, method of delivery and group or individual delivery)
► Comparator information (eg, setting, content, intervention duration and frequency, profession delivering the intervention, method of delivery and group or individual delivery)
► Outcomes (eg, dietary outcome(s) studied, how they were measured, duration of follow-up, statistical analysis and intervention effect sizes)
► BCTs, as reported or described
► Social media-enabled function or features (eg, gamification of interface, online community and chatbots)

The following items will be extracted from the eligible qualitative studies:

► General information (eg, study authors, publication year, country and funding source)

► Study details (eg, study aim, study design, randomisation method, blinding and allocation concealment)
► Participant information (eg, demographic characteristics, sample size and health conditions)
► Intervention information (eg, setting, content, BCTs, intervention duration and frequency, method of delivery and group or individual delivery)
► Comparator information, if there are (eg, setting, content, intervention duration and frequency, profession delivering the intervention, method of delivery and group or individual delivery)

## Risk of bias in individual studies

The quality assessment will be conducted at both the study and outcome levels, using the Mixed Methods Appraisal Tool[48] for its ability to evaluate diverse study designs. This tool incorporates a spectrum of criteria tailored specifically for the analysis of quantitative, qualitative and mixed-method research articles. To ensure a rigorous and unbiased assessment, two reviewers will independently appraise the articles and discuss any discrepancies to reach a consensus.

The certainty of evidence for each study will also be assessed using the GRADE criteria[49] to identify the strengths and limitations of the evidence, and it informs the conclusions and recommendations that can be made based on the review findings.

## Data synthesis

A 'convergent segregated approach' to data synthesis will be applied.[50] We will (1) organise and analyse the qualitative and quantitative data separately, (2) integrate the data using a triangulation method and (3) conduct narrative synthesis based on research questions and objectives to describe the findings from both the quantitative and qualitative data.

For the quantitative studies reporting on changes in diet, eating behaviours and patterns, if there are sufficient estimates (at least three studies) from independent studies, we will conduct a random-effect meta-analysis to pool and synthesise the results.[51] Based on this model, estimates of standardised mean differences for continuous outcomes and risk ratios for binary outcomes, as well as 95% CIs for each outcome, will be provided. Where appropriate, publication bias will also be statistically assessed using funnel plots and Egger's test.

However, if there are limited data, or when the studies included in the review have significant heterogeneity in terms of design, population, interventions or outcomes, Synthesis Without Meta-Analysis (SWiM) will be used.[52] SWiM involves synthesising and summarising the findings from individual studies in a narrative or descriptive manner, rather than through statistical pooling of data. This will ensure transparency and reproducibility in the synthesis process. Where appropriate, a Harvest Plot[53] will be developed and used to visually present quantitative findings.

For the qualitative studies, thematic analysis[54] will be conducted to identify common themes and patterns in the qualitative data.[54] This will allow us to identify a deeper understanding of the experiences, perspectives and motivations of young people participating in the intervention. This will involve reading and coding all text and tables labelled results (or equivalent), identifying themes and summarising the findings. We will focus on identifying participants' preferences, expectations and feelings about the intervention and facilitators and barriers to intervention use.

Finally, triangulation[55] will be used to bring together the results of the qualitative and quantitative syntheses to provide a comprehensive and integrated understanding of the impact of social media interventions on promoting healthy eating behaviours among young people. We will present the results of each synthesis on one page and analyse the convergence, complementarity or discrepancy of the findings. Additionally, a joint display[56] will be employed, visually representing the integrated results to offer a more holistic perspective on the outcomes.

### Assessment of heterogeneity
Heterogeneity between the studies in effect measures will be assessed using the $I^2$ statistic. We will consider an $I^2$ value >50% as indicative of substantial heterogeneity. If there are sufficient data to warrant it, we will further conduct sensitivity analyses to explore heterogeneity based on study quality and/or study characteristics (see subgroup analyses). Analyses would be undertaken in R.

### Sensitivity analysis
Suitable sensitivity analysis will be informed by the decisions made during the review process and based on consultation with the review team. Sensitivity analysis may be conducted to explore how *risk of bias* (as indexed by the RoB 2.0 tool)[57] and participant characteristics affect estimates.

### Analysis of subgroups
In the presence of sufficient data, subgroup analysis of prespecified groupings will be performed for the following study characteristics:
- ► Population characteristics (eg, age and gender)
- ► Intervention type (eg, diet only vs exercise only vs diet and exercise combination, including vs excluding psychological support)
- ► Intervention duration (eg, 1 day, 4 weeks, 12 weeks.)
- ► Frequency and duration of follow-up[15]
- ► Intervention approach (individual, segmentation and parental involvement)[15 38]
- ► Intervention content (weight loss, mental health and combined)
- ► Intervention delivery method (via social media only, social media combined with text messages and social media combined with in-person contact/workshops)
- ► Reported or coded BCTs (eg, education, goal setting and action planning) using the BCT Taxonomy v1 classification framework[58]
- ► Reported functions and features of social media intervention (eg, gamification (rewards), communities and groups)

**Acknowledgements** We would like to extend our gratitude to the librarian Veronica Phillips from the Cambridge University Medical Library for their invaluable support in designing the literature search strategy. We also acknowledge and thank Jessica Kohli for her assistance with the screening process of this review.

**Contributors** HT served as the lead author of the study protocol, responsible for conceptualising and designing the study, and drafting the manuscript. AS and ALA provided guidance and feedback throughout the entire process, helping to refine the research question and improve the study design and manuscript. EvS provided input into the development of the protocol and contributed to RoB assessment. MS helped screen abstracts and full texts and refine inclusion criteria. All authors have read and approved the final protocol.

**Funding** This research did not receive specific funding from an agency in the public, commercial or not-for-profit sectors. ALA, AS and EvS are funded by the Medical Research Council (grants MC_UU_00006/6 and MC_UU_00006/5).

**Competing interests** None declared.

**Patient and public involvement** Patients and/or the public were not involved in the design, or conduct, or reporting, or dissemination plans of this research.

**Patient consent for publication** Not applicable.

**Ethics approval** The systematic review does not require ethics clearance since published studies with non-identifiable data will be used. The results of the systematic review will be disseminated via publication in a peer-reviewed journal as well as through conference presentations.

**Provenance and peer review** Not commissioned; externally peer reviewed.

**ORCID iD**
Hao Tang http://orcid.org/0000-0003-3002-1374

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
