## [Reviewer comments · BMJ Open]

ARTICLE DETAILS

TITLE (PROVISIONAL)	The impact of social media interventions on eating behaviours and diet in adolescents and young adults: a mixed methods systematic review protocol
AUTHORS	Tang, Hao; Spreckley, Marie; van Sluijs, Esther; Ahern, Amy; Smith, Andrea

VERSION 1 – REVIEW

REVIEWER	Garcia del Castillo, Jose Antonio Universitas Miguel Hernández
REVIEW RETURNED	15-Jan-2024

GENERAL COMMENTS	The approach of this study is relevant and very interesting. The problems related to the use of social networks in food is of paramount importance in order to tackle this problem in preventive programmes.
--

REVIEWER	Chung, Alicia NYU Grossman School of Medicine
REVIEW RETURNED	23-Jan-2024

GENERAL COMMENTS	The protocol is very thorough, on a very fascinating topic. My concern is that while the statistical analysis is very robust, the final set of studies that actually result at the end of the review may be very small, and not have the statistical to power to run many of the statistical analysis proposed. This may be especially challenging, given the topic of the review includes eating behaviors and diet, which may include eating disorders, and not just healthy eating. This was evident by the scoping review I published on the topic, Chung A, Vieira D, Donley T, Tan N, Jean-Louis G, Kiely Gouley K, Seixas A Adolescent Peer Influence on Eating Behaviors via Social Media: Scoping Review J Med Internet Res 2021;23(6):e19697 URL: https://www.jmir.org/2021/6/e19697 DOI: 10.2196/19697 I look forward to reading the manuscript on the results of the systematic review.
---

VERSION 1 – AUTHOR RESPONSE

Reviewer #1

Dr. Jose Antonio Garcia del Castillo, Universitas Miguel Hernández

Comments to the Author:

The approach of this study is relevant and very interesting. The problems related to the use of social networks in food is of paramount importance in order to tackle this problem in preventive programmes.

Author response: Thank you for your positive feedback.

Reviewer #2

Dr. Alicia Chung, NYU Grossman School of Medicine

Comments to the Author

The protocol is very thorough, on a very fascinating topic. My concern is that while the statistical analysis is very robust, the final set of studies that actually result at the end of the review may be very small, and not have the statistical to power to run many of the statistical analysis proposed. This may be especially challenging, given the topic of the review includes eating behaviors and diet, which may include eating disorders, and not just healthy eating. This was evident by the scoping review I published on the topic, Chung A, Vieira D, Donley T, Tan N, Jean-Louis G, Kiely Gouley K, Seixas A Adolescent Peer Influence on Eating Behaviors via Social Media: Scoping Review

J Med Internet Res 2021;23(6):e19697

URL: <https://www.jmir.org/2021/6/e19697>

DOI: 10.2196/19697

I look forward to reading the manuscript on the results of the systematic review.

Author response: Thank you for your valuable feedback on our manuscript protocol. We appreciate your insights on the statistical analysis and the potential challenges with the final set of studies. We will address these considerations in our systematic review findings and discuss the implications of any limitations on the statistical power.

Your scoping review on adolescent peer influence on eating behaviours is enlightening and will inform our study further. We look forward to sharing the results of our systematic review with you and the wider research community soon.